# A stitch in time: The importance of water and sanitation services (WSS) infrastructure maintenance for cholera risk. A geospatial analysis in Harare, Zimbabwe

**Sophie Ayling**[1]*, **Sveta Milusheva**[2], **Faith Maidei Kashangura**[3], **Yi Rong Hoo**[1], **Hugh Sturrock**[4], **George Joseph**[1]

1 Water Global Practice, World Bank Group, Washington DC, United States of America, 2 Development Impact Evaluation Unit (DIME), World Bank Group, Washington DC, United States of America, 3 Department of Works, City Planning and Development, Harare, Zimbabwe, 4 Lytchett House, Poole, Dorset, United Kingdom

* sayling@worldbank.org, sophie2ayling@gmail.com

**Data Availability Statement:** All public data is being made available on the GitHub link https://github.com/dime-worldbank/geospatial-cholera-

## Abstract

Understanding the factors associated with cholera outbreaks is an integral part of designing better approaches to mitigate their impact. Using a rich set of georeferenced case data from the cholera epidemic that occurred in Harare from September 2018 to January 2019, we apply spatio-temporal modelling to better understand how the outbreak unfolded and the factors associated with higher risk of being a reported case. Using Call Detail Records (CDR) to estimate weekly population movement of the community throughout the city, results suggest that broader human movement (not limited to infected agents) helps to explain some of the spatio-temporal patterns of cases observed. In addition, results highlight a number of socio-demographic risk factors and suggest that there is a relationship between cholera risk and water infrastructure. The analysis shows that populations living close to the sewer network, with high access to piped water are associated with at higher risk. One possible explanation for this observation is that sewer bursts led to the contamination of the piped water network. This could have turned access to piped water, usually assumed to be associated with reduced cholera risk, into a risk factor itself. Such events highlight the importance of maintenance in the provision of SDG improved water and sanitation infrastructure.

## Author summary

Gaining an understanding of the spatial and temporal dynamics of prior epidemics, and the factors that may have driven transmission, helps public health agencies identify opportunities for preventing further outbreaks. This study outlines an analysis of the cholera epidemic that occurred in Harare, Zimbabwe, between September 2018 – January 2019. The study makes use of data on individual cases, where the household location and time of each case are known, to explore factors associated with cholera transmission and to retrospectively generate detailed maps of weekly transmission risk across the epidemic. As

modelling-zim/tree/main/data and the World Bank Microdata Catalog https://microdata.worldbank.org/index.php/catalog/?page=1&ps=15. Access channels for proprietary data are listed in S2 Table 'Data Access Information'. Email addresses are cohweb@hararecity.co.zw, callcentre@zinwa.co.zw and pr@mohcc.gov.zw'.

**Funding:** We would like to acknowledge with thanks the financial support of the Global Water Security and Sanitation Partnership (GWSP) of the World Bank and the ieConnect for Impact Program funded with UK aid from the UK government (Foreign Commonwealth & Development Office) in completing this work. Awards were received by GJ and SM respectively. The funders had no role in study design, data collection and analysis, decision to publish, or preparation of the manuscript. The following are the links to their respective websites: GWSP https://www.worldbank.org/en/programs/global-water-security-sanitation-partnership ieConnect https://www.worldbank.org/en/research/dime/brief/transport FCDO https://www.gov.uk/government/organisations/foreign-commonwealth-development-office.

**Competing interests:** The authors have declared that no competing interests exist.

part of these analyses, Call Detail Records (CDR) from a major mobile phone network provider were analysed to help estimate and include information on human movement throughout the epidemic period. Results show several potential risk factors, and suggest that populations living in close proximity to the sewer network with high access to piped water, were at higher risk. One possible explanation for this observation is that sewer bursts led to the contamination of the piped water network, with a higher concentration of pathogens at the point of origin of the burst. This could have turned access to piped water, usually assumed to be associated with reduced cholera risk, into a risk factor itself.

## Introduction

Cholera is caused by the etiological agent Vibrio cholerae O1 and O139 serogroups and is transmitted through the faecal-oral route by consuming food or water containing the bacteria [1]. The World Health Organization (WHO)'s Global Task Force on Cholera Control views cholera not just as a global threat to public health, but also an indicator of poverty and a mark of inequity [2]. Since 1970, cholera has become endemic in many countries of sub-Saharan Africa [3] with annual Case Fatality Ratios (CFRs) of 1.6%—the highest in the world [4]. While several Oral Cholera Vaccines (OCVs) now exist, they only protect against cholera infection for up to two years, and they address the symptoms and not the cause. Large outbreaks continue to plague several countries, especially in sub-Saharan Africa. Without addressing the pathways to contamination in these low-income settings with inadequate infrastructure, outbreaks will continue to be a risk.

Pathways for transmission can include contaminated water sources, both for drinking and domestic use [5–8], poor sanitation [9–11]; contaminated food, poor personal hygiene [11] and poor drainage [12,13]. Furthermore, spatio-temporal analyses have demonstrated how cases can cluster among close contacts and therefore be more prevalent in dense living environments where an uncontaminated water supply is not readily available for drinking, washing or cooking [14]. These same pathways can also lead to other water-borne diseases such as typhoid (from the etiological agent *Salmonella enterica* serovar Typhimurium [15], making it vital to understand which pathways are most important in a given setting in order to target resources effectively to reduce the spread of these diseases. Additionally, since cholera is a water-borne disease, the focus in terms of risk factors is usually on infrastructure and water sources, yet another important factor is mobility. If infected individuals travel from an area with a cholera outbreak to an area without any cases and the water/sewer networks are susceptible to contamination, the traveller could contaminate the water in this new location and lead to the spread of the disease through host shedding into water supplies that are subsequently ingested [16,17]. Studying the role of mobility could be important for helping to contain spread from host shedding [18–20].

In particular, outbreaks in emerging nation contexts are usually attributed to where households have low access to 'improved' water and sanitation infrastructure such as networked water and sewer systems ('improved' water and sanitation is as defined by the WHO Joint Monitoring program) [3,21]. However, poorly maintained, or operated network infrastructure can also fail to bring about the protection from cholera that it may seem to promise. There are examples from developed and middle-income countries where poor operation of sewerage infrastructure can lead to health and environmental hazards [22–24].

An understanding of the spatial distribution of cholera risk, and the factors driving risk, can support decision making around longer-term interventions to reduce the likelihood of outbreaks. Especially in cities with mixed service provision—where households may rely on piped water and point sources for drinking water, and on-site sanitation or sewer networks for

sanitation—there are a variety of potential improvements that could be made. These range from investing in repair or construction of network infrastructure (water or sewer pipes), to ensuring better access to protected point sources (boreholes, protected wells) or improving the quality of on-site sanitation (such as pit latrines or septic tanks). While in general, improving access to piped water is considered an improvement in infrastructure with public health benefits, piped water networks can become health hazards if not properly maintained. For example, in Zambia, Ashraf et al. found that service interruptions were associated with an increased incidence of diarrheal disease, upper respiratory infections and typhoid fever [25]. Equally, sewer networks are considered to be the top of the sanitation service ladder according to the Joint Monitoring Program of the WHO [26] with risks from onsite facilities more well-documented in the literature [27]. Nevertheless, when bursts or leakages are not properly addressed, liquid sewage can potentially be even more hazardous than waste from onsite sanitation facilities [28].

This paper brings together a rich set of geolocated indicators on water and sanitation, population characteristics and mobility, that jointly are rarely available in a lower-income setting. These indicators are used to conduct a geo-spatial analysis to explore the role of water and sanitation infrastructure on cholera risk. The study focuses on Harare, Zimbabwe, using the cholera outbreak of September–December 2018 as a case study. We use spatial modelling, detailed data on infrastructure and mobility, and georeferenced case data to identify factors associated with cholera risk. Based on this correlational analysis, we discuss different scenarios of infrastructural improvements and how they may be associated with reduction in relative risk when targeting by ward across the city.

## Methods

### Ethics statement

This study was approved by the Medical Research Council of Zimbabwe. The approval number is MRCZ/E/302 and it was assessed as exempt. All data obtained and herein reported were anonymized prior to analysis.

### The study area

Zimbabwe is a landlocked Sub-Saharan African country with a population of just under 15 million people [29] It is divided into 60 districts and bordered by Zambia, Mozambique, Botswana, and South Africa. Its capital, on which this analysis is focused, is Harare, a city with a population of 1.5 million inhabitants [30] and 46 wards. Zimbabwe has experienced cholera outbreaks dating back to 1971 with the worst occurring in 2008–2009, which resulted in 4,288 reported deaths over 60 of its 62 districts [31]. The focus of this paper is a more localized but severe outbreak which took place between September and December 2018 and was concentrated in the capital Harare. Of almost ten thousand cases, 93% (9,755) were reported in Harare, resulting in 46 deaths. According to the latest Demographic and Health Survey (DHS) data from 2015, 30% of Harare's population is connected to an existing sewer network [32], compared to just 12% in neighbouring country capital Lusaka, Zambia from the same year of data collection [33].

All the data detailed in the paragraphs below was cleared for use by the Medical Research Council of Zimbabwe (MRC-Z).

### Case data

Georeferenced cholera case data were available for the Harare area collected by the Ministry of Health and Child Care (MoHCC) following the World Health Organization (WHO) guidelines. The first case was recorded on September 4th, 2018, and the last on January 9th, 2019.

The Zimbabwean national surveillance system criteria for case diagnosis describes a suspected case as any person aged 2 years or more with acute watery diarrhoea, with or without vomiting. It describes a confirmed case as a suspected case in which V. cholerae serogroups O1 or O139 was isolated from a stool sample [34]. There were 216 confirmed cases in the data that we received and used in our analysis.

Date of diagnosis, gender, age, and geo-location of the case's dwelling were collected. The geo-location of the case's dwelling was calculated as part of a comprehensive initiative on the part of the Harare City Council Department of Works to match home addresses reported by households to the city cadastral map. For spatio-temporal analyses, cases were aggregated across seven-day bins from September 1st, 2018. Age and gender of the patients are not considered in this analysis. There was a total of 9,977 cases in the line list. 87 of the observations either had no addresses and therefore could not be geocoded or were duplicate entries and were removed before sharing with the research team. Therefore, ultimately, 9,890 cases were included in the analysis after cleaning. Of these, 216 were confirmed while the remainder were suspected. Though the original case data cannot be disclosed for privacy reasons, dummy case data is available in the GitHub repository (https://github.com/dime-worldbank/geospatial-cholera-modelling-zim/tree/main/data) for replication purposes.

## Population characteristics data

Population density data were available from Facebook Humanitarian Data Exchange (HDX) [35]. Small Area Estimates (SAEs) for poverty produced by the World Bank were used for ward area estimates of poverty [36]. Household access to water, sanitation and hand washing infrastructure were taken from the DHS 2015. This was available at the cluster level for 60 clusters for Harare and the surrounding area, and 44 inside the city. There was an average of 26 households per cluster for the sample areas. To generate continuous estimates of sanitation risk, a beta Generalized Additive Model (GAM) with linear associations with population density, poverty, proximity to the water network and river and night light intensity plus a spatial term modelled using a bivariate smooth on latitude and longitude, was fit to sanitation risk scores of 0–1 (rescaled to 0.01–0.99 for the purpose of modelling). Sanitation risk scores were first calculated according to the Joint Monitoring Program (JMP) SDG Improved Sanitation ladder [26]. The rankings were equally spaced with the lowest risk of 0 being those households who reported having an unshared sewered connection OR an unshared VIP latrine, pit latrine with slab or septic tank. A risk level of 1 was given to households who had either of the former two technology types but with a shared sanitation facility due to risk of contamination; those who reported unimproved sanitation facilities had a risk score of 4, and those who reported open defecation had the highest risk score of 5. These categorical values were then normalized to create 166 unique values and rescaled to lie between 0.01 and 0.99 for modelling purposes. Modelling was conducted using a Beta regression resulting in a continuous sanitation risk scores ranged from 0.08–4.88 (IQR 1.519 3.065). Separately the proportion with access to piped water and unimproved water was modelled and predicted across the city using a multinomial GAM with a spatial term where the outcome was one of three classes: unimproved, improved, and piped water. This was not done by ranking but instead by creating a dummy variable on improved or piped water. Because the proportion of the population with unimproved water was very low, this resulted in high correlation (0.94) of the proportion with piped and protected water sources. To avoid issues of collinearity in subsequent spatial modelling of cholera risk, therefore, only proportion with piped and unimproved water sources were retained. This data and corresponding processing code is available on the GitHub repository referenced in the case data section.

## Water and sanitation infrastructure and environment data

Data on water and sanitation infrastructure in the city was collected from a variety of sources. The water and sanitation network shape files for the city were obtained with the support of the City of Harare's (CoH) Water Department and Department of Works. The shapefiles were generated as part of an initiative that was started in 2012 to digitize all hard copies of CoH Water Department's water and sanitation network structures. The process was completed in 2019 during which time no major infrastructure upgrades took place. Small edits were manually added in the digitized files. This data is proprietary. Please see S2 Table for data access information.

Complaints data were digitized for 7,179 water network breakage reports and 24,713 sewer burst location reports from around the city from December 2017—December 2018 with the support of City of Harare's Department of Works. These complaints were logged in a complaints book with the address location of the burst and then digitized by referring to the cadastral maps. Though the data was digitized up to December 2019, for the analysis we divided the dataset into pre- the Jan 2019 outbreak and afterwards. We used the data for the pre- outbreak period in the analysis. Along with water quality samples for 23 sampling points from each month for August 2018 –February 2019, this data is also proprietary and can be made available on request using the same communication channel as for the WSS infrastructure data. Administrative boundaries for water supply administrative areas (known as DMAs) (165 suburbs), and the location of 362 markets (to include flea markets, fruit and veg stands, arts and crafts and tyre sales) and public facilities (such as car parks, home industries and nurseries) were also collected and digitized by City of Harare's GIS department. The latter has been made authorized for public release and is available on the GitHub repository.

Data on 1,848 private borehole locations was provided by Manyame, Upper Mazowe and Nyagui Sub-catchment area operators, of which 1,721 were located in Harare. This data is collected by the water resources [37] authority ZIMWA as individuals drilling private boreholes are required to gain a permit from the authority by law since 2012. Some boreholes which have been illegally sunk would be missing from this list. Finally, a groundwater vulnerability index was developed by Ziva, a Zimbabwean Data & Knowledge Management Consultant Company, based on a combination of geological factors (fault index, soil permeability, depth to groundwater, topographic wetness index and fissure flow). Each of these factors, both individually and in conjunction with one another will affect the ability and speed of contaminated water flow below ground. Given that many households do rely on groundwater for drawing water through wells when the network supply is unreliable, ordinarily we would expect the groundwater vulnerability to have implications for cholera risk. If the boreholes are drilled deeper than the flow of contaminated water, the risk of contamination may be reduced if the well is considered 'protected' due to the lining of the well. However, all the factors mentioned will come into play to determine both how deep contamination may infiltrate the groundwater, and how likely it will be to spread across distances. More information on groundwater flows can be found in Focazio et al. 2002 [38]. The combined index has been made available on the github repository.

## Population mobility data

We use anonymous and aggregated Call Detail Record (CDR) data provided by two Mobile Network Operators in Zimbabwe covering the period of the outbreak (September 2018-February 2019) and processed by a team at the Development Impact Evaluation Unit (DIME), World Bank following a data sharing agreement between the two parties. Together, these operators made up 76% of the market share in 2019 based on data from TeleGeography.

Anonymous CDR data contain information on the location of the nearest cell phone tower through which any calls or texts made or received are routed. By comparing the locations of where these calls or texts are placed for the same anonymized Subscriber Identity Module (SIM) over time, it is possible to estimate movement patterns for any SIM down to the spatial resolution of the cell phone tower catchment area [39–41]. The derivative product is property of World Bank. A summary of all data sources described here, and their access levels is provided in the S1 and S2 Tables.

## Spatial modelling

One common method to model point event data, such as locations of disease cases, is to aggregate over an arbitrarily sized grid [40].This results in a count per grid cell, which can then be modelled using a Poisson regression. However, given that the size of the grid used to aggregate points is essentially arbitrary, results are conditional on the choice of grid size. While in theory it would be possible to decrease the size of the grid used (i.e., increase the spatial resolution of the grid), this can quickly become computationally expensive, particularly if the data in question are to be modelled over space and time, making it necessary to aggregate spatially for each period.

A more suitable approach is to apply Poisson point process models which allow models to be fit to the point event data directly. Poisson point process models are suitable for modelling the distribution of points in space and time and have been used in a number of settings including modelling disease case data, species distributions [42] and locations of storm peaks [43].

An important aspect of the models of cholera cases is the inherent spatio-temporal variation in disease risk which should be accounted for as part of any analysis. Log Gaussian Cox processes offer one way to include a spatio-temporal component to the model which can be computationally expensive to fit, although more efficient methods based on integrated nested Laplace approximation and stochastic partial differential equation models have been proposed [44].

Similar to the approach taken by Youngman and Economou [45], here we fit Poisson point process models using a GAM framework [44]. GAMs provide a flexible and efficient way to fit regression models and, particularly those involving spatial and spatio-temporal data alongside covariate effects. The approach tries to estimate the relationship between covariates, including the spatio-temporal term, and the intensity surface that gave rise to the cases. A standard way to estimate this relationship is using numerical integration at observation (i.e., case) point locations combined with a set of quadrature points [46], which are essentially evenly distributed points across the spatio-temporal domain of interest. A critical aspect of this approach is assigning quadrature weights to each point which then act as an offset term in the regression model. For most Poisson point process modelling (e.g., species distribution modelling using presence only data) this would be the area of the domain represented by the quadrature point. Where quadrature points are distributed over a regular lattice, this weight would be equivalent to the spatial resolution of the cells formed by the lattice. When modelling disease cases, however, the quadrature weights (offset) should reflect the population in the area represented by the quadrature point. Here we set the quadrature weight equal to the population in the corresponding lattice cell, excluding any cells in which cases occurred. Weights for cases were set as the population in the cell in which the case arose. The number of quadrature points per seven-day period was set to be 4,000, which inspection of model fit statistics suggested sufficient.

To account for residual spatio-temporal effects, the GAM used to model intensity included a tensor product between a bivariate Gaussian process smooth on latitude and longitude and a cubic regression spline on time. While covariate effects could be modelled using non-linear

splines using this approach, here we opted to fit linear effects to simplify interpretation of associations.

S1 Table lists details of the covariates examined in the modelling process.

The covariates explored were selected based on plausible relationships with cholera risk. We include water infrastructure due to the well-known association between contaminated drinking water and water used for domestic purposes with cholera risk [5–7]. Though piped water is considered to be safer by World Health Organization standards, it depends on a continuous service through sealed pipes and with water treatment included. This is not always the case with poor quality piped infrastructure [25]. Sanitation infrastructure and sanitation facilities are also included due to their known association with cholera risk. For onsite facilities contamination can occur through unlined or poorly constructed pit latrines which seep content into the groundwater supply. Thus, groundwater vulnerability is also included in the model because it feeds into the many borewells in the city [47]. Meanwhile, sewer networks where breakages or overflows are common are also known to contaminate the water bodies they discharge into [48]. There are also illegal connections that can cause negative pressures. However, those working closely with the utility report that intermittent supply is the main cause of pressure surges, compromising system integrity. Vandalism outstanding bursts and aging infrastructure also serve to exacerbate maintenance challenges. From sewer pipes this may result in groundwater contamination when pipes are buried, or surface water if untreated sewage is discharged directly into streams or rivers [9–11]. We include the market and public facility locations as well as the population density data due to the pathways to contamination that have been identified in contaminated food and poor personal hygiene [11]. In addition, cholera transmission has been associated with case clusters among close contacts respectively [14]. We finally include an indicator that accounts for mobility by calculating the expected imported cases that come in from other areas (Fig 1). Emigration could lead to the generation of additional cholera cases in an area, given research showing that population mobility can lead to spread of cholera to new areas [17–20]. While distance to small and large sized markets was also considered, these were excluded from the modelling process given high correlation (greater than 0.8) with other covariates. All covariates were produced at, or resampled to, approximately 100m resolution (Fig 2).

To integrate human movement into the model, we use a methodology drawn from Milusheva et al. to calculate the risk of imported malaria [39]. For every individual/SIM we calculated the probability that an individual traveling to a destination became infected in the last ten

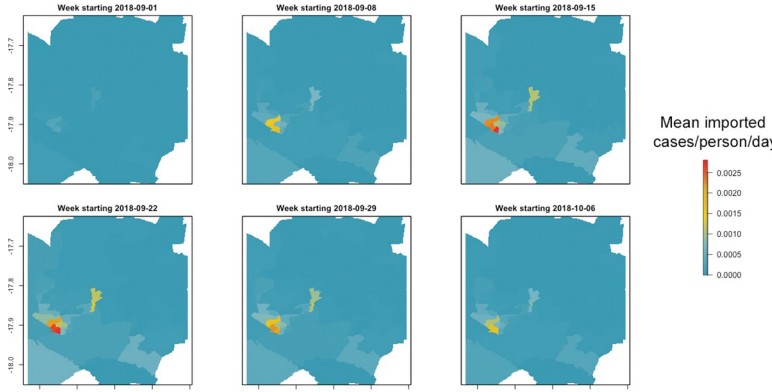

**Fig 1. Estimated incidence of imported cholera cases for the first 6 weeks of the outbreak in Harare.** Map made with R package Leaflet v2.1.1 and map data from OpenStreetMap.

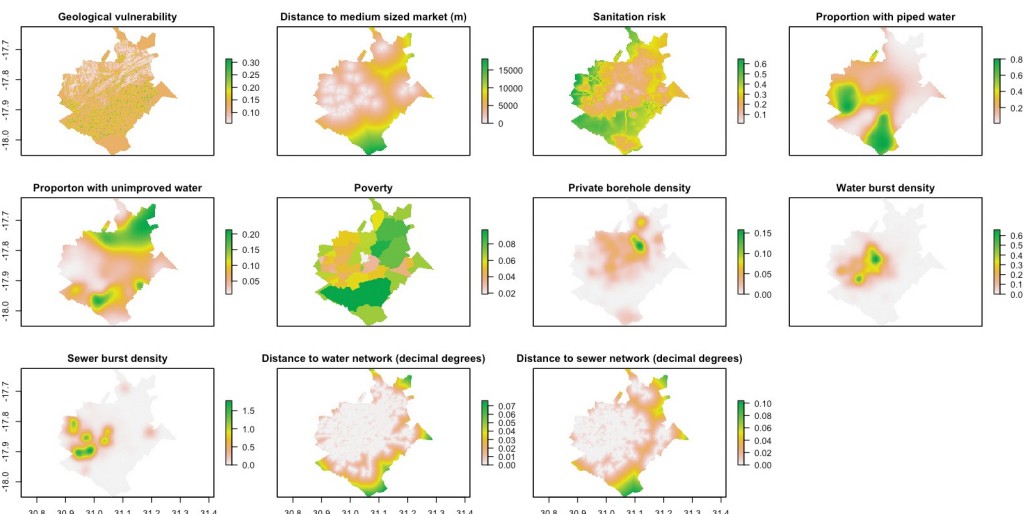

**Fig 2. Covariates included in the point process modelling.** Map made with R package Leaflet v2.1.1 and map data from OpenStreetMap.

days. This calculation was based on the 1) locations the individual spent time in during those ten days, 2) duration of time spent in each location and 3) local level of risk at each location (estimated by calculating observed weekly incidence per tower catchment). We then sum probabilities across all individuals entering a given destination ward in each week to calculate the total expected number of imported cases of cholera that week.

This resulted in an estimate of the incidence of 'imported' cases per ward for each week. This was rasterized to the same resolution and extent as the other covariates and included in the model as a dynamic covariate lagged by 1 week (i.e., incidence of imported cases in the week prior). Given there is an incubation period and potentially additional wait time before a new case is recorded in a facility, using this lag was meant to capture new cases that may have been generated because of infected individuals that have entered the area and "imported" cholera into the given destination.

These static and dynamic covariates were included in the model as linear effects. A backwards stepwise approach was taken with covariates retained in the model if their inclusion led to a decrease in AIC of greater than two. In addition, to examine the hypothesis that faults in the sewer network may have led to contamination of the piped water network, we included an interaction between the proportion of households with piped water and distance to the sewer network.

Model fit was evaluated by plotting the observed number of cases per time period against the number predicted by the model. In addition, we constructed a hex grid of approximately 1km width and compared the number of observed cases across time periods per grid cell to the number predicted.

## Results

Fig 3 shows the distribution of cases temporally and spatially. Cases were concentrated towards the first few weeks in the south-west area of the city in the Glen View and Budiriro neighbourhoods.

All covariates except for water burst density were retained in the final model, including the interaction between proportion with piped water and distance to the sewer network. Table 1 shows the coefficients for the final model. The final model had an AIC value of 56034. The same model without lag incidence of imported cases had an AIC of 56219 and a spatio-

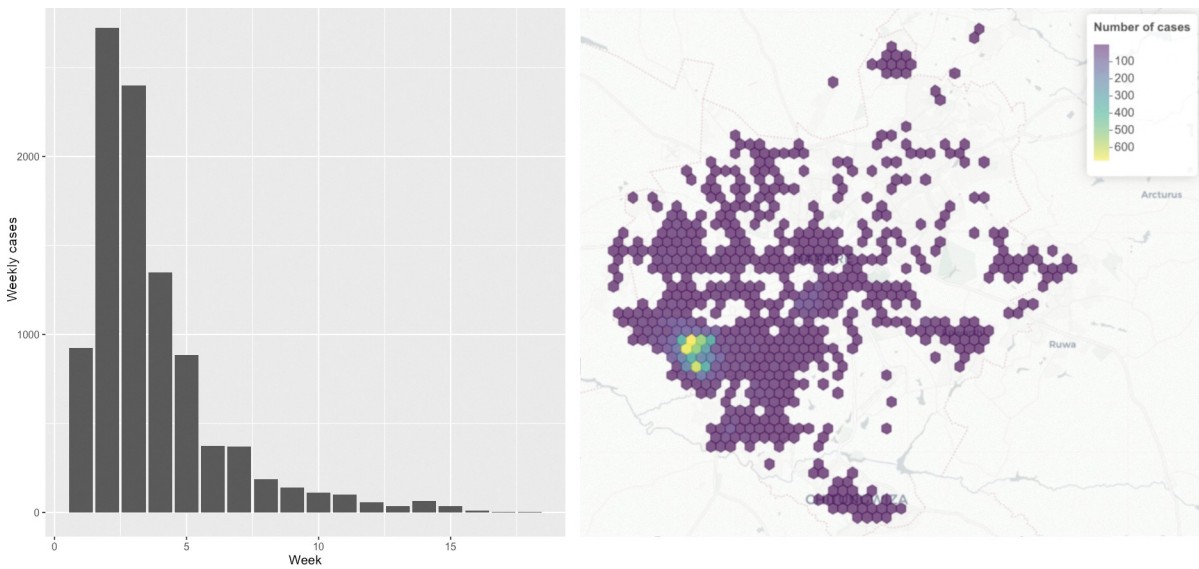

**Fig 3. Distribution of cholera cases across Harare temporally by week (left) and spatially across all weeks (right).** Basemap attribution: CARTO (basemaps linked here), OpenMapTiles, OpenStreetMap contributors (license linked here).

temporal only model (i.e., with no covariates) had an AIC of 57943. This indicates that the addition of covariates and lag incidence improved the model fit. Improvement of model fitting was made using Hastie et al. as a reference [49].

Model validation showed that the fitted values showed good alignment with the observed numbers of cases, both per week and per 1km hex grid cell (Figs 4 and S1).

The models suggested that sanitation risk, poverty, sewer burst density and incidence of imported cases were all risk factors for cholera (Table 1). In contrast, increasing geological vulnerability, distance to medium sized market, proportion with unimproved or other water source, private borehole density and distance to water network were all negatively associated with risk. To aid interpretation, Table 2 shows the relative risk per covariate using a scaled approach. Rather than showing the relative risk for every unit increase in the covariate, which

**Table 1. Results from the point process modelling of cholera risk in Harare.** Coefficients represent the difference in the log of the expected rates for a unit increase in the covariate.

| Term | Coefficient (Rate ratio) | 95% CI | p-value |
|---|---|---|---|
| Intercept | -5.7930 | -8.17, -3.41 | <0.0001 |
| Geological vulnerability | -2.9880 | -3.75, -2.23 | <0.0001 |
| Distance to medium sized market (m) | -0.0005 | -0.000643, -0.000399 | < 2e-16 |
| Sanitation risk | 4.2810 | 3.53, 5.03 | < 2e-16 |
| Proportion with piped water | 2.2090 | -3.96, 8.38 | 0.4831 |
| Proportion with unimproved/other water | -81.5300 | -100, -62.7 | < 2e-16 |
| Poverty | 14.3600 | 10.3, 18.5 | <0.0001 |
| Private borehole density | -36.2900 | -61.6, -11.0 | 0.0049 |
| Sewer burst density | 3.1790 | 2.38, 3.98 | <0.0001 |
| Distance to water network (decimal degrees) | -199.3000 | -231, -167 | < 2e-16 |
| Distance to sewer network (decimal degrees) | 20.3800 | -15.4, 56.2 | 0.2645 |
| Lag incidence of imported cases | 246.7000 | 176, 318 | <0.0001 |
| Proportion with piped water * Distance to sewer network (decimal degrees) | -729.1000 | -830, -628 | < 2e-16 |

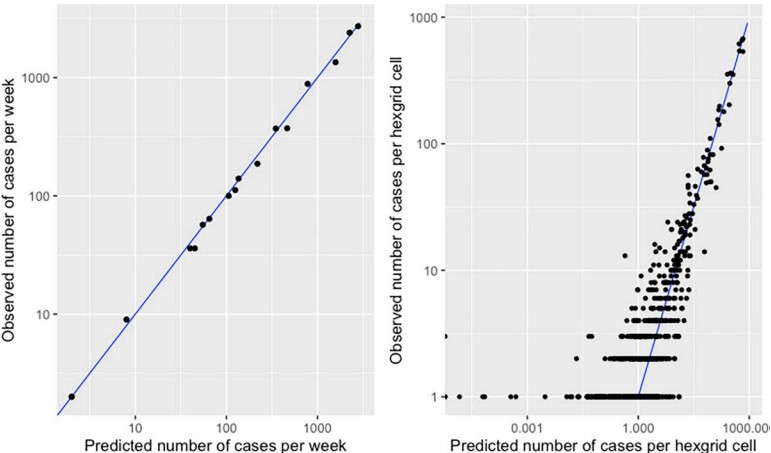

**Fig 4.** Observed versus predicted numbers of cases per week (left) and per ~1km hex grid cell (right).

is dependent on the scale of the covariate, here we calculate the relative risk between covariate strata, where strata are defined by dividing each covariate into 10 equal width strata across the range of covariate values found in populated grid cells. For example, if the values in populated grid cells of a given covariate range from zero-two hundred, the relative risk is shown for every increase of twenty in the covariate. This shows that differences in geological vulnerability, poverty and lag incidence of imported cases were associated with relatively small changes in risk with relative risks per incremental stratum of 0.92, 1.12 and 1.07 respectively. In contrast, other covariates, particularly proportion with unimproved water, sewer burst density and distance to water network were associated with larger differences in risk.

The models also suggested that there was an interaction between the proportion of households with piped water and distance to the sewer network. Fig 5 describes this interaction, showing the relationship between the proportion of households with piped water and cholera risk at different distances from the sewer network when all other covariates are set to their mean. This shows that adjacent to the sewer network (at zero km), risk is slightly higher, the higher the proportion of households with piped water. However, this relationship switches direction in areas greater than two km from the sewer network, where the higher the

**Table 2. Relative risk of moving into the next stratum of a covariate (excluding interaction between distance to sewer and proportion with piped water) where strata are formed by dividing the covariate values (in populated grid cells) into ten equal width strata.** For example, if the values in populated grid cells of a given covariate ranges from zero–two hundred, the relative risk is shown for every increase of twenty in the covariate.

| Term | Range of covariate value across population of Harare | Width of covariate stratum (range / 10) | Relative risk per incremental stratum |
| --- | --- | --- | --- |
| Geological vulnerability | 0.056–0.312 | 0.026 | 0.92 |
| Distance to medium sized market | 0–18,006 | 1,800 | 0.39 |
| Sanitation risk | 0.014–0.692 | 0.068 | 1.34 |
| Proportion with unimproved/other water | 0.009–0.212 | 0.02 | 0.19 |
| Poverty | 0.019–0.097 | 0.008 | 1.12 |
| Private borehole density | 0–0.158 | 0.016 | 0.56 |
| Sewer burst density | 0–1.778 | 0.178 | 1.76 |
| Distance to water network (decimal degrees) | 0–0.077 | 0.01 | 0.21 |
| Lag incidence of imported cases | 0–0.0028 | 0.0028 | 1.07 |

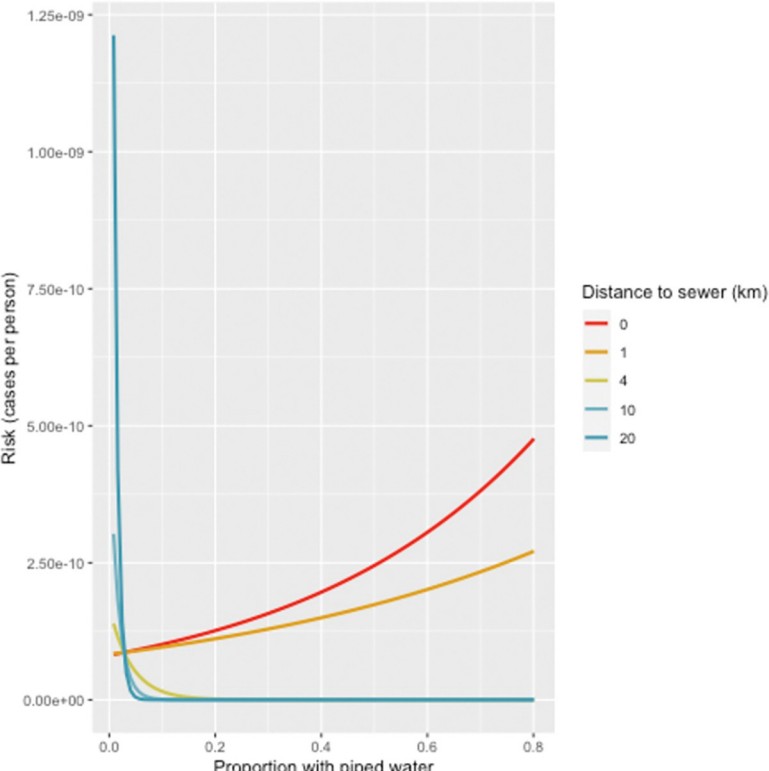

**Fig 5. The effect of proportion with piped water at different distances to the sewer network.** The y axis represents the risk (cases per person) when holding all other covariates at their mean value. Higher values indicate higher cholera risk.

proportion of households with piped water, the lower the risk of cholera. (A log visualization of this relationship is provided in S3 Fig).

Fig 6 shows the predicted incidence at 100m resolution across Harare for the first six weeks in which 88.6% of cases were recorded. Incidence is markedly higher towards the southwest part of the city around Glen View and Budiriro neighbourhoods.

## Discussion

There are a few key findings that come out of this research and have important policy implications. As demonstrated by the coefficients in Table 2, the models found an association between sanitation risk, poverty, sewer burst density and incidence of imported cases with increased cholera risk. Meanwhile, increasing geological vulnerability, distance to a medium sized market, proportion with unimproved or other water source, private borehole density and distance to water network were associated with reduced cholera risk. While the correlation of poor sanitation with cholera risk [12] is well known in the literature, including in the context of a previous cholera outbreak in Zimbabwe [50], the rich geolocated data we have compiled allows us to identify and associate specific types of sanitation infrastructure. Our model results also align with previous literature demonstrating an association with poverty [51]. Associated factors with poverty and market size are often identified in the literature such as population density, overcrowding and low educational status [52,53]https://www.zotero.org/google-docs/?1kGWLd. The association with reduced risk of proximity to the water network also complements similar findings in Lusaka, Zambia [12].

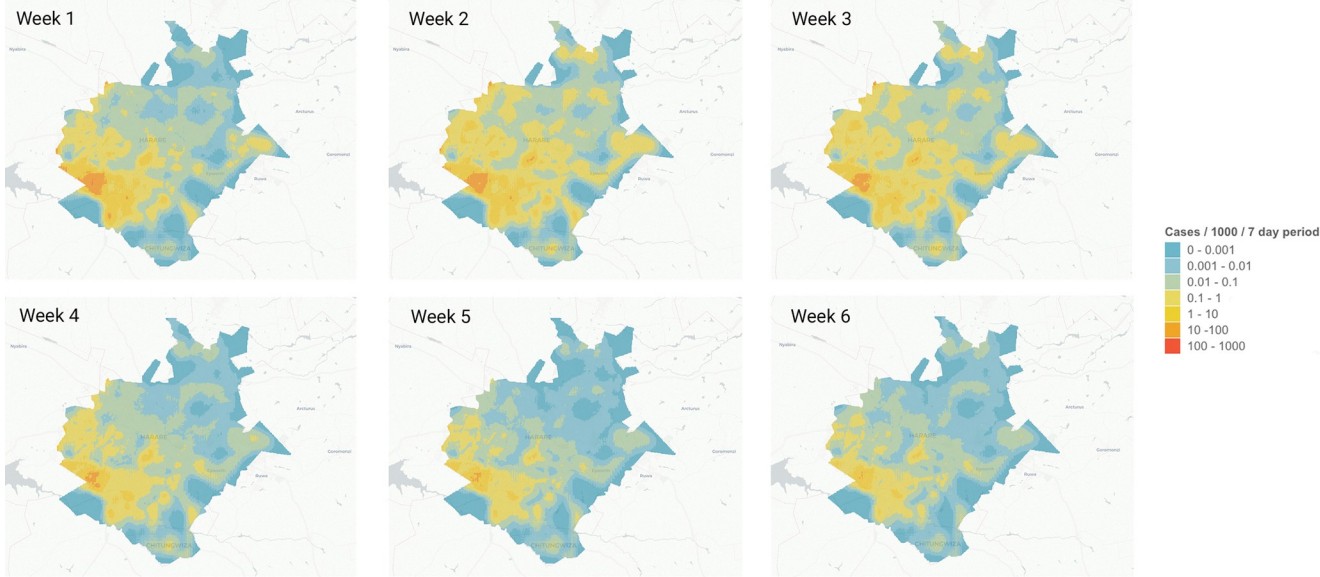

**Fig 6. Predicted incidence of cholera across Harare for the first 6 weeks of the outbreak (September 1ˢᵗ–October 13ᵗʰ, 2018).** Note the log-scaled colour palette. Basemap attribution: CARTO (basemaps linked here), OpenMapTiles, OpenStreetMap contributors (license linked here).

A counterintuitive finding from this work is that the proportion of those with unimproved water source was associated with reduced cholera risk. The model suggests that in Harare, there was an association between the combination of the proportion of households with piped water and distance to the sewer network, with increased cholera risk. In other words, it suggests that the closer to the sewer network, the greater the association with increased cholera risk, for households with piped water. However, this relationship switches direction in areas greater than two km from the sewer network, where having piped water is associated with lower risk of cholera. It is unclear as to why those with a higher proportion of unimproved water are associated with lower cholera risk and may be explained by contamination of the piped water network which temporarily makes unimproved water sources relatively safer. In addition, there is undoubtedly error in the predicted proportion with unimproved water as these data are based on a small number of obfuscated DHS data points which could lead to a spurious relationship. The DHS has a policy of randomly displacing the GPS coordinates of households to protect the privacy of respondents by between zero and two km. Similarly, ordinarily we would expect that the relationship with geological vulnerability to be associated with increased cholera risk, especially if sewage or OSS effluent is proximate to unprotected drinking water sources. Therefore, in this case, the reasons for a lack of a negative association require further exploration.

The most accurate way of testing whether the piped water closer to the sewer network is more contaminated would be through water quality testing data at different points along the network. Although water quality testing data was provided for this analysis, there were insufficient data points to be able to make such a comparison. In the absence of such data, this analysis can still provide some useful insights into spatial targeting of investments. As shown in Fig 1, sewer bursts are spatially very concentrated in a few areas. Therefore, it could be possible to target sewer maintenance investments in these few areas, and particularly in areas which are closer to the piped water network. Assuming these areas represent locations of the sewer network more vulnerable to future bursts, doing so would thereby offer health benefits in terms of reducing cholera risk, while minimizing costs since it would require investments in only a few

areas. Such targeting is illustrated by example scenarios provided in Annex 2 on how the effectiveness of different targeting schemes could be calculated. Though costing data was not available for us to extend the analysis in that direction, by virtue of targeting a few wards, rather than the whole city, the study offers policymakers some indication on the best use of resources.

We identify that spatially, the risk of sewer bursts on this occasion was concentrated in five wards in the southwest of the city (S2 Fig). It is likely that sewer bursts do not occur at random but are spatially correlated due to underlying weaknesses in the network in those areas. Therefore, targeting the five wards where the majority of sewer bursts were reported could be an efficient way to reduce cholera risk relative to repairing sewer bursts across the city. This is supported by our data, that a high density of sewer bursts was positively associated with cholera cases. Using the data from the bursts reported and analysed, there appeared to be relatively little increased benefit (in terms of reducing cholera risk) in investing in repairing sewer bursts throughout the city as opposed to just focusing on the five wards mentioned. The efficiency and cost effectiveness of reducing cholera risk is therefore much greater when concentrating on areas that fall within the top 25th percentile of sewer bursts. It is also more efficient to provide piped water to within 500 meters of the existing network, rather than to extend access to the whole city when considering their relative impact on reducing the number of cholera cases.

The association between sewer bursts and cholera risk is important given that according to the Joint Monitoring Program of the WHO, who set the standards for sanitation globally, sewered sanitation sits at the top of the improved sanitation ladder. However, it serves to highlight the importance of proper maintenance of existing sanitation networks rather than necessarily emphasizing new construction. This is particularly important in a context where the institutional capacity and financial resources to maintain large scale sanitation systems may be lacking.

An important limitation of the study is that the analysis presented is only able to draw correlations and associations, rather than drawing causal inferences given we are not able to use some form of exogenous variation to identify the causal impact of individual factors. Instead, we look at the relative predictive power of a small set of indicators, which can help to provide insights into factors that may be important for policy makers to consider in further analysis. This clearly has important implications when interpreting the scenario analysis. Secondly, when considering data quality and reliability, we must consider that a number of layers used in the analysis, such as the proportion of households with access to piped water, were generated using DHS data, which have been spatially obfuscated. As a result, this may have introduced error into the layer, which could have reduced the accuracy of predicted associations with cases. Sewer and water burst density was based on spatio-temporally referenced complaints data, which is obviously a proxy measure with a number of reported and recorded issues. Data from direct inspections of sewer and water networks would have been a more reliable measure but such data was not available to the researchers. Finally, the case data is based on the line list, which though geo-referenced to households, does rely largely on suspected rather than confirmed case data. Although this data collection method does not deviate from the WHO GTFCC guidelines, having a larger proportion of verified cholera cases may have changed the distribution of the data used in the analysis. A recommendation for further outbreaks may be to increase the proportion of confirmed cases with respect to suspected.

The focus of this analysis has been to better understand spatial variation in risk of cholera, using Harare, Zimbabwe as a case study that could help to provide insights for low-income urban settings, and to study what factors might be associated with cholera risk. A natural extension of the work would be to focus on some of the less intuitive associations and identify settings where it may be possible to study them in a causal framework. This could then lead to

more direct policy implications of factors that need to be tackled, which could be combined with cost estimates to compare different targeted interventions. Furthermore, if additional data were available on factors driving sewer bursts, the analysis could be enhanced to have predictive capacity to reduce the risk of future outbreaks. Above all, this analysis shows the importance of emphasizing maintenance above and beyond new construction, especially when thinking of extending an already poorly maintained system.

## Supporting information

**S1 Fig. Observed and predicted numbers of cases over a ~1km hexgrid.** Map made with R package Leaflet v2.1.1 and map data from OpenStreetMap.
(TIF)

**S2 Fig.** Left—The five wards in which targeting interventions would prevent the largest number of cases for the scenarios A–no sewer bursts, B–no sewer bursts within 200m of the water line, C—No sewer bursts in high (median) density burst areas and D—no sewer bursts in highest (75th percentile) density burst areas. Right–the cumulative reduction (%) in cases achieved, and population targeted, from targeting wards in order of priority. Base map attribution: CARTO, OpenMapTiles, OpenStreetMap contributors.
(TIF)

**S3 Fig. The effect of proportion with piped water at different distances to the sewer network.** The y axis represents the risk (log of risk) when holding all other covariates at their mean value. Higher values indicate higher cholera risk.
(TIF)

**S1 Table. Details of covariates in the modelling process.**
(XLSX)

**S2 Table. Data access information.**
(XLSX)

## Acknowledgments

The authors also gratefully acknowledge the cooperation of officials from City of Harare (CoH) Water and GIS Departments, in particular Toine Ramaker from VEI, Dr. Manangazira from the Ministry of Health, Zimbabwe; in country counterparts at the World Health Organization (WHO); and the Country Management Unit (CMU) of the World Bank in Zimbabwe, in particular then Country Manager Mukami Kariuki, Senior Operations Officer Tonderai Fadzai Naome Mukonoweshuro and Health Specialist Consultant Chenjerai Sisimayi for facilitating contact with counterparts and local dissemination of findings. From the Water GP, we also received invaluable support from Anna Cestari, Senior Water Resource Management Specialist, Ireen Mangoro, Water Consultant, Chris Heymans, Senior Water Supply and Sanitation Specialist. Without their support this work would not have been possible.

The findings, interpretations and conclusions expressed in this paper do not necessarily reflect the views of the World Bank, the Executive Directors of the World Bank, or the governments whom they represent. The World Bank does not guarantee the accuracy of the data included in this work.

Official delimitation of areas and borders do not necessarily reflect the official position of the World Bank Group. Country borders or names do not necessarily reflect the World Bank Group's official position. These maps are for illustrative purposes and do not imply the

expression of any opinion on the part of the World Bank, concerning the legal status of any country or territory or concerning the delimitation of frontiers or boundaries.

## Author Contributions

**Conceptualization:** George Joseph, Sophie Ayling, Hugh Sturrock.

**Data curation:** Sophie Ayling, Faith Maidei Kashangura, Yi Rong Hoo.

**Formal analysis:** Hugh Sturrock.

**Investigation:** Yi Rong Hoo.

**Methodology:** George Joseph, Sveta Milusheva, Hugh Sturrock.

**Supervision:** George Joseph.

**Writing – original draft:** Sophie Ayling, Sveta Milusheva, Hugh Sturrock.

**Writing – review & editing:** Sophie Ayling, Sveta Milusheva, Faith Maidei Kashangura, Yi Rong Hoo, Hugh Sturrock, George Joseph.

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
