## [Decision Letter · Decision Letter 0]

22 Aug 2022

Dear Ms Ayling,

Thank you very much for submitting your manuscript "The importance of Maintenance: Geospatial Analysis of Cholera Risk and Water and Sanitation Infrastructure in Harare, Zimbabwe" for consideration at PLOS Neglected Tropical Diseases. As with all papers reviewed by the journal, your manuscript was reviewed by members of the editorial board and by several independent reviewers. In light of the reviews (below this email), we would like to invite the resubmission of a significantly-revised version that takes into account the reviewers' comments. 

Your paper was peer reviewed by three experts in the field. Two felt the manuscript needed only minor revisions, whereas the third reviewer felt strongly that some major changes were needed. Please respond point by point to the issues raised by the reviewers and indicate changes you made to the manuscript to address each issue. If you disagree with the reviewer's opinion on a particular point, you may indicate this and give reasons why as well.

We cannot make any decision about publication until we have seen the revised manuscript and your response to the reviewers' comments. Your revised manuscript is also likely to be sent to reviewers for further evaluation.

Sincerely,

Jeffrey H Withey

Guest Editor

Elsio Wunder Jr

Section Editor

Your paper was peer reviewed by three experts in the field. Two felt the manuscript needed only minor revisions, whereas the third reviewer felt strongly that some major changes were needed. Please respond point by point to the issues raised by the reviewers and indicate changes you made to the manuscript to address each issue. If you disagree with the reviewer's opinion on a particular point, you may indicate this and give reasons why as well.

Reviewer's Responses to Questions

**Key Review Criteria Required for Acceptance?**

**Methods**

-Are the objectives of the study clearly articulated with a clear testable hypothesis stated?

-Is the study design appropriate to address the stated objectives?

-Is the population clearly described and appropriate for the hypothesis being tested?

-Is the sample size sufficient to ensure adequate power to address the hypothesis being tested?

-Were correct statistical analysis used to support conclusions?

-Are there concerns about ethical or regulatory requirements being met?

Reviewer #1: (No Response)

Reviewer #2: Yes but further review from health economist is suggested

Reviewer #3: - The statistical approach used is not well described and no link to source code is provided. The authors cite a number of papers when describing their treatment of CDR data and the Poisson point process model but a formal description of what exactly they are doing is needed for me (and presumably most readers). Ideally the authors would also publish their data and code to allow for their results to be reproduced.

- It isn’t clear how all the covariates in the models were decided upon. Was there a DAG or some theoretical model illustrating the hypothesis of associations to make sure simply throwing them all in and selecting based on AIC didn’t lead to unintentional confounding (e.g., collider bias)? The approach used seems reasonable if the aim is prediction but less ideal for inference about individual covariates (and scenarios). 

- To what spatial scale were pipe bursts etc digitised? Modeling is done at a pretty small scale and it would be good to make clear that there is reasonable certainty that digitisation of the water and sanitation infrastructure data are done at an appropriate scale. The authors should be even more clear in methods that this is just reports of pipe bursts but not necessarily confirmed pipe bursts. 

- Are sewer bursts time varying covariates in the model? If so, I think it would be helpful to understand the associations across different lags. 

- It would be helpful to describe which health facilities were included in the line list. Is there any risk that the locations of the primary cholera treatment centers influenced the observed spatial distribution of cases? Showing the health centers on a map along with cumulative incidence would be useful. 

- As far as I can tell this is based entirely on suspected cholera case counts. The authors should be more clear about this. Further the authors discussion confirmation but don’t say the method used nor whether these data were used in anyway. 

- It appears these data are from the MoHCC linelist. It would be helpful for the authors to better describe the at which admin level the cases were georeferenced to. Were there any missing data? How were these handled? 

- For the WHO case definition provided, are areas where cholera is not known to be occurring neighbourhoods? Or is Harare treated as a unit and the suspected case definition is really just ≥ 2 y/o with AWD (except for the first case)?

- Water and Sanitation Infrastructure Data: These data are from what period of time? Same as the outbreak or something longer term?

- It would be helpful to understand the marketshare of the two providers of CDR data to help the reader understand how representative these might be. 

- The following data availability statement is not consistent with PLOS’s data policy “Data is stored on the World Bank OneDrive. It can be accessed and shared in consultation with Harare City Council's Water utility. Cholera case data would have to be consulted with the Zimbabwean Ministry of Health and the WHO.” Data are supposed to be made open except in special circumstances (e.g., when they are identifiable).

**Results**

-Does the analysis presented match the analysis plan?

-Are the results clearly and completely presented?

-Are the figures (Tables, Images) of sufficient quality for clarity?

Reviewer #1: (No Response)

Reviewer #2: Yes

Reviewer #3: - The authors find that almost all coefficients in their regression model do not have confidence intervals spanning the null. This is surprising to me given how rarely these variables have been associated with cholera incidence in the past and how small effect sizes tend to be. While I could be wrong (and can’t fully assess given the lack of detailed methods), it seems like this is an artefact of the methods used which don’t appropriately account for correlated data. 

- In addition, it would be helpful for the authors to report interpretable coefficients (presumably exponentiated and perhaps with additional transformations). Hard to know how important these seemingly large effect sizes are. 

- It is not clear to me how to interpret the results suggesting increases (e.g., 1e16!) in incidence from scenarios with sewer access provided to the entire city and those with piped water….

- “Surprising findings from this work include the fact that proportion of those with unimproved water source is a protective factor and that increases in geological vulnerability is also a protective factor. The former can be explained by the proximity of piped water sources to sewers with a high number and density of bursts.” This may be the case but it also suggests that adjusting for bursts as done in these models is insufficient.

**Conclusions**

-Are the conclusions supported by the data presented?

-Are the limitations of analysis clearly described?

-Do the authors discuss how these data can be helpful to advance our understanding of the topic under study?

-Is public health relevance addressed?

Reviewer #1: (No Response)

Reviewer #2: Yes

Reviewer #3: (No Response)

**Editorial and Data Presentation Modifications?**

Reviewer #1: (No Response)

Reviewer #2: Overall, this is a well written manuscript.

Reviewer #3: - Figure 3: y-axis label should be weekly suspected cases, not total cases, correct?

- Under figure 3 it says “Table X shows”

- “An additional penalty term was added to each covariate to allow non-significant variables to essentially be selected out of the model.” - What exactly did you do here?

- “The results of the scenario modeling are shown in table x.”

- Figure 5: I suggest not presenting the numbers on the log scale. I don’t think many will exponentiate in their heads.

**Summary and General Comments**

Reviewer #1: (No Response)

Reviewer #2: This paper uses geo-spatial analysis to explore the role of water and sanitation infrastructure on

cholera risk taking case study of Harare, Zimbabwe. The findings of study are interesting. However, some findings such as proportion of those with unimproved water source is a protective factor and that increases in

geological vulnerability is also a protective factor. The interpretation and justification of these findings is not convincing and need careful analysis and interpretation.

Reviewer #3: The manuscript “The importance of Maintenance: Geospatial Analysis of Cholera Risk and Water and Sanitation Infrastructure in Harare, Zimbabwe” presents results of a statistical spatio-temporal analysis exploring the associations between various potential risk factors related to the built environment, natural environment and socio-demographic factors and cholera incidence in Harare at a fine spatial scale. They then use the fitted regression model to explore the predicted incidence when setting covariates to different levels. The authors find associations between cholera incidence and several potential risk factors (far more than most studies find) and suggest that eliminating pipe bursts would have led to large reductions in cholera incidence. I think this is an interesting story but the authors do not explain their methods sufficiently to understand what exactly they are doing and from what I do understand, seem to be drawing very causal conclusions when at best they are estimating (potentially biased) associations. This reads much more like a report than a manuscript and If this were to be published I would expect much more details, more sensitivity analyses and more restrained interpretation of the results.

PLOS authors have the option to publish the peer review history of their article (what does this mean?). If published, this will include your full peer review and any attached files.

Reviewer #1: No

Reviewer #2: Yes: DR Meghnath Dhimal

Reviewer #3: No
---

## [Decision Letter · Decision Letter 1]

31 Jan 2023

Dear Ms Ayling,

Thank you very much for submitting your manuscript "A Stitch in Time: The Importance of Water and Sanitation Services (WSS) Infrastructure Maintenance on Cholera Risk. A Geospatial Analysis in Harare, Zimbabwe" for consideration at PLOS Neglected Tropical Diseases. As with all papers reviewed by the journal, your manuscript was reviewed by members of the editorial board and by several independent reviewers. In light of the reviews (below this email), we would like to invite the resubmission of a significantly-revised version that takes into account the reviewers' comments. 

The reviewers were supportive of your revised manuscript but still had some serious concerns that need to be addressed before we can consider publication. Please respond again to each of the reviewers comments, point by point, and include this with the revised manuscript.

We cannot make any decision about publication until we have seen the revised manuscript and your response to the reviewers' comments. Your revised manuscript is also likely to be sent to reviewers for further evaluation.

Sincerely,

Elsio Wunder Jr, D.V.M., Ph.D.

Section Editor

Elsio Wunder Jr

Section Editor

The reviewers were supportive of your revised manuscript but still had some serious concerns that need to be addressed before we can consider publication. Please respond again to each of the reviewers comments, point by point, and include this with the revised manuscript.

Reviewer's Responses to Questions

**Key Review Criteria Required for Acceptance?**

**Methods**

-Are the objectives of the study clearly articulated with a clear testable hypothesis stated?

-Is the study design appropriate to address the stated objectives?

-Is the population clearly described and appropriate for the hypothesis being tested?

-Is the sample size sufficient to ensure adequate power to address the hypothesis being tested?

-Were correct statistical analysis used to support conclusions?

-Are there concerns about ethical or regulatory requirements being met?

Reviewer #1: (No Response)

Reviewer #3: (No Response)

**Results**

-Does the analysis presented match the analysis plan?

-Are the results clearly and completely presented?

-Are the figures (Tables, Images) of sufficient quality for clarity?

Reviewer #1: (No Response)

Reviewer #3: (No Response)

**Conclusions**

-Are the conclusions supported by the data presented?

-Are the limitations of analysis clearly described?

-Do the authors discuss how these data can be helpful to advance our understanding of the topic under study?

-Is public health relevance addressed?

Reviewer #1: (No Response)

Reviewer #3: (No Response)

**Editorial and Data Presentation Modifications?**

Reviewer #1: (No Response)

Reviewer #3: (No Response)

**Summary and General Comments**

Reviewer #1: (No Response)

Reviewer #3: Thank you for a chance to re-review this manuscript. I appreciate the authors responses to the reviewer and editor comments and believe the paper is much clearer now. However, I have several pending and new comments that need attention:

- The authors do not explain why the data cannot be shared to be consistent with the PLOS Data Policy. Pointing to someone in the Ministry of Land for non-identifiable data is not a sustainable solution to this as a single person rarely will stay in this type of position and there is no clear criteria by which requests for data will be assessed. This includes data on pipe bursts, admin zone boundaries etc. I do not expect GPS points of case households to be shared but many other data sources should be shared. Furthermore, points can be jittered or aggregated (as I doubt the address data were exact). This is really important to allow for reproducibility of results. 

- In my previous review I had asked about the rational for using different covariates and the authors explained it was based on previous hypotheses and literature (which is now clarified in the text). One missing bit is the geologic vulnerability. What was the thought behind this? In the discussion the authors suggest that they don’t have a plausible mechanism for this relationship so I am not sure I understand why they decided to include it in the first place. 

- In response to my previous question about case definitions the authors responded with: “For the WHO case definition provided, are areas where cholera is not known to be occurring neighbourhoods? Or is Harare treated as a unit and the suspected case definition is really just ≥ 2 y/o with AWD (except for the first case)?” The authors simply cut and paste the GTFCC reccomended case definitions. This does not answer the question of how this was done in Harare. In the methods section the authors should be explicit about what the Ministry of Health used as case definitions in this outbreak exactly not just what WHO/GTFCC recommends. Would also be useful to mention how many cholera cases were culture confirmed during this outbreak.

- The authors created a sanitation risk score from 0-1 based on the JMP sanitation ladder but it is not clear from the Methods how exactly this was done. Is each rung of the ladder equally spaced? Why was this done rather than a multinomial like what was done for water indicators? Please update methods to clarify.

- The coefficients in Table 1 remain uninterpretable to the general reader. I realise it is difficult to interpret these but this is indeed a public health journal and general readers will need to be able to map these numbers to some interpretation (especially if you chose to focus on them as a table). I encourage the authors to spend some time thinking about how these could be transformed into interpretable results that could possibly be used in decision making and policy. 

Minor:

- Pg 6 - “Finally, a groundwater vulnerability index was developed by Ziva, based on a combination of geologica..” What is Ziva?

- The track changes version of the manuscript uploaded does not highlight all changes made and it is really hard to assess differences. 

- Figure 4: I suggest transforming the axes given that most of the data are less than 100. With current scale its very hard to see the majority of error. 

- No reference to GitHub page within manuscript. Please add this to the text.

 - Provide inferred flux matrices in GitHub repo. 

- The authors responded to my query about the time frame of sewer burst data but did not add this to the paper. Please add that the sewer burst data used in the model was from Dec 2017 through Dec 2018. 

- Clarify in limitations that these were based on suspected, not confirmed cholera cases and how this could impact results.

- Can the authors comment on whether illegal connections to the piped water system may compromise its integrity (sometime creating negative pressure)? I know this is an issue in some places but not sure about Harare. 

- Pg 9, “Covariates were retained in the model if their inclusion led to a decrease in AIC of >2.” Is this through some forward/backward selection process? Please specify in text.

- Double check text that there are no references to the now deleted scenario analyses.

PLOS authors have the option to publish the peer review history of their article (what does this mean?). If published, this will include your full peer review and any attached files.

Reviewer #1: No

Reviewer #3: No
---

## [Editor Report · Decision Letter 2]

3 May 2023

Dear Ms Ayling,

We are pleased to inform you that your manuscript 'A Stitch in Time: The Importance of Water and Sanitation Services (WSS) Infrastructure Maintenance on Cholera Risk. A Geospatial Analysis in Harare, Zimbabwe' has been provisionally accepted for publication in PLOS Neglected Tropical Diseases.

Best regards,

Elsio Wunder Jr, D.V.M., Ph.D.

Section Editor

Jeffrey Withey

Academic Editor

---

## [Editor Report · Acceptance letter]

30 May 2023

Dear Ms Ayling,

We are delighted to inform you that your manuscript, "A Stitch in Time: The Importance of Water and Sanitation Services (WSS) Infrastructure Maintenance for Cholera Risk. A Geospatial Analysis in Harare, Zimbabwe," has been formally accepted for publication in PLOS Neglected Tropical Diseases.

Best regards,

Shaden Kamhawi

co-Editor-in-Chief

Paul Brindley

co-Editor-in-Chief
